# The Maastricht Acquisition Platform for Studying Mechanisms of Cell–Matrix Crosstalk (MAPEX): An Interdisciplinary and Systems Approach towards Understanding Thoracic Aortic Disease

**DOI:** 10.3390/biomedicines11082095

**Published:** 2023-07-25

**Authors:** Berta H. Ganizada, Koen D. Reesink, Shaiv Parikh, Mitch J. F. G. Ramaekers, Asim C. Akbulut, Pepijn J. M. H. Saraber, Gijs P. Debeij, Armand M. Jaminon, Ehsan Natour, Roberto Lorusso, Joachim E. Wildberger, Barend Mees, Geert Willem Schurink, Michael J. Jacobs, Jack Cleutjens, Ingrid Krapels, Alexander Gombert, Jos G. Maessen, Ryan Accord, Tammo Delhaas, Simon Schalla, Leon J. Schurgers, Elham Bidar

**Affiliations:** 1Departments of Cardiothoracic Surgery, CARIM School for Cardiovascular Diseases, Heart and Vascular Center, Maastricht University Medical Center (MUMC+), 6229 ER Maastricht, The Netherlands; berta.ganizada@mumc.nl (B.H.G.);; 2Department of Biochemistry, CARIM School for Cardiovascular Diseases, Heart and Vascular Center, Maastricht University Medical Center (MUMC+), 6229 ER Maastricht, The Netherlands; 3Department of Biomedical Engineering, CARIM School for Cardiovascular Diseases, Heart and Vascular Center, Maastricht University Medical Center (MUMC+), 6229 ER Maastricht, The Netherlands; 4Department of Radiology and Nuclear Medicine, CARIM School for Cardiovascular Diseases, Heart and Vascular Center, Maastricht University Medical Center (MUMC+), 6229 ER Maastricht, The Netherlands; 5Department of Cardiology, CARIM School for Cardiovascular Diseases, Heart and Vascular Center, Maastricht University Medical Center (MUMC+), 6229 ER Maastricht, The Netherlands; 6Stem Cell Research University Maastricht Facility, 6229 ER Maastricht, The Netherlands; 7Department of Vascular Surgery, CARIM School for Cardiovascular Diseases, Heart and Vascular Center, Maastricht University Medical Center (MUMC+), 6229 ER Maastricht, The Netherlands; 8Department of Pathology, CARIM School for Cardiovascular Diseases, Heart and Vascular Center, Maastricht University Medical Center (MUMC+), 6229 ER Maastricht, The Netherlands; 9Department of Clinical Genetics, CARIM School for Cardiovascular Diseases, Heart and Vascular Center, Maastricht University Medical Center (MUMC+), 6229 ER Maastricht, The Netherlands; 10Department of Vascular Surgery, University Hospital RWTH Aachen, 52074 Aachen, Germany; 11Department of Cardiothoracic Surgery, Center for Congenital Heart Diseases, University Medical Center Groningen, 9713 GZ Groningen, The Netherlands; 12Institute of Experimental Medicine and Systems Biology, RWTH Aachen University, 52074 Aachen, Germany

**Keywords:** aortic dissection, tissue biomechanics, mechanobiology, smooth muscle contractility, elastin degradation, connective tissue disorder, collagen turnover

## Abstract

Current management guidelines for ascending thoracic aortic aneurysms (aTAA) recommend intervention once ascending or sinus diameter reaches 5–5.5 cm or shows a growth rate of >0.5 cm/year estimated from echo/CT/MRI. However, many aTAA dissections (aTAAD) occur in vessels with diameters below the surgical intervention threshold of <55 mm. Moreover, during aTAA repair surgeons observe and experience considerable variations in tissue strength, thickness, and stiffness that appear not fully explained by patient risk factors. To improve the understanding of aTAA pathophysiology, we established a multi-disciplinary research infrastructure: The Maastricht acquisition platform for studying mechanisms of tissue–cell crosstalk (MAPEX). The explicit scientific focus of the platform is on the dynamic interactions between vascular smooth muscle cells and extracellular matrix (i.e., cell–matrix crosstalk), which play an essential role in aortic wall mechanical homeostasis. Accordingly, we consider pathophysiological influences of wall shear stress, wall stress, and smooth muscle cell phenotypic diversity and modulation. Co-registrations of hemodynamics and deep phenotyping at the histological and cell biology level are key innovations of our platform and are critical for understanding aneurysm formation and dissection at a fundamental level. The MAPEX platform enables the interpretation of the data in a well-defined clinical context and therefore has real potential for narrowing existing knowledge gaps. A better understanding of aortic mechanical homeostasis and its derangement may ultimately improve diagnostic and prognostic possibilities to identify and treat symptomatic and asymptomatic patients with existing and developing aneurysms.

## 1. Introduction

Ascending thoracic aortic aneurysm (aTAA) represents a significant cause of morbidity and mortality worldwide. The prevalence of aTAA in the general population is 0.16% to 0.34%, with an incidence of 10 cases per 100,000 patient-years [1]. Asserting these rates is a challenge, as patients with aTAA are mostly asymptomatic until aortic dissection or rupture occurs [2]. Thoracic aortic diameter is normally 3.2 cm and is defined as an aneurysm when the diameter exceeds 4 cm because this presents a z-score > 2 and shows a >6000 times increased relative risk of dissection [3]. aTAA of >6 cm has an annual incidence of rupture or dissection of up to 7% [4]. Therefore, an increased aortic diameter is currently the core risk factor for dissection. Current European and American management guidelines for degenerative aTAA recommend surgical intervention once ascending or sinus diameter reaches 5–5.5 cm or shows a growth rate of >0.5 cm/year, as assessed by computed tomography (CT), echocardiography, and magnetic resonance imaging (MRI) [5]. It should be noted that between modality differences in thresholds are to be considered.

For high-risk patients with a genetic predisposition or bicuspid valve morphology, lower thresholds (4.0–5.0 cm diameter or growth rates > 0.3 cm/year) are considered [6,7]. Around 20% of aTAA are associated with a positive family history or identified genetic mutation [8], as involved in Marfan, Loeys–Dietz, and vascular Ehlers–Danlos syndromes.

However, up to 96% of aTAA dissections (aTAAD) occur in vessels with diameters below the surgical intervention threshold [9]. Also, most patients with ascending aortic aneurysms of 4 cm show little to no further growth during prolonged follow-up [10]. The sole use of diameter or dilatation rate oversimplifies the complexity of aTAA formation, as they do not consider body size, wall composition, and hemodynamic loading of the vessel [11]. Beyond patient risk factors, surgeons observe and experience unexplained differences in aneurysm tissue strength, thickness, and stiffness during aTAA repair procedures.

Beyond diameter [12] and genetic susceptibility (from genes such as FBN-1, TGFB(R), MYH11 and ACTA2), clinical translational research has focused on blood biomarkers involved in aTAA formation, e.g., the elastin degradation and fibrillin fragmentation products desmosine (DES) and isodesmosine (ISDES) [13], as well as physical properties, such as wall strain, aortic elongation and volume [9]. Aortic wall strain can be assessed in vivo, though subject to modality-specific limitations (e.g., resolution, loading conditions), through echocardiography [14], computed tomography [15], and MRI [16,17], and using tensile testing ex vivo [18]. Abnormal wall shear stress and strain show relevant associations with aTAA development [19].

Considering the above, there is a clear need for a platform of research approaches and infrastructures, to enable the integration of clinical, genetic, mechanical, and biological concepts and data, to further our understanding of ascending thoracic aortic disease.

In the present paper, we introduce and describe the MAPEX platform as initiated at our institutes. Because the approach and implementation of MAPEX are dictated by existing and integrative concepts around mechanical homeostasis and mechanobiology, we will describe the scientific underpinning and main research question in the next section.

## 2. Scientific Underpinning and Research Question

### 2.1. Loss of Mechanical Homeostasis

Mechanical homeostasis describes the active processes of biological tissue to maintain form and mechanical properties, such as strength, resilience, and elasticity [20]. It is illustrative to consider the mechanical demands on a normal ascending aorta. With each heartbeat, the vessel undergoes a mechanical stretch of about 10–20%. Assuming a life course duration of 80 years, the aorta must accommodate such elastic stretches about 2.5 billion times, invoking a tremendous feat of resilience and/or adaptation. Particularly, if one considers the longevity of an artificial/manufactured rubber tube, it appears evident that indeed active processes should be central to mechanical homeostasis.

Accordingly, we approach aTAA(D) from a pathophysiological perspective, focusing on where and how homeostatic processes derange or get disrupted towards pathology.

Much of the current understanding of aTAA pathophysiology comes from histopathological studies of disease requiring cardiac surgery, where aTAA has been shown to associate with elastin degradation, cystic medial necrosis, and vascular smooth muscle cell (VSMC) loss within the tunica media of the aortic wall [12,21]. Acquired and hereditary conditions exacerbate or accelerate these processes [22]. In addition to loss, VSMCs have been found to exhibit a synthetic rather than contractile phenotype, which is a hallmark of arterial remodelling [23]. A switch, or modulation, of VSMCs towards a synthetic phenotype is characterised by decreased expression of a contractile protein (i.e., α-actin and myosin) and increased production of matrix metalloproteinases (MMPs). Moreover, synthetic VSMCs secrete extracellular vesicles that may promote local calcification and inflammation [24,25]. While these biological findings and insights are associated with aneurysmal pathology, their interrelatedness and involvement in the loss of mechanical homeostasis remain to be elucidated [26].

Because the dynamic interactions between VSMCs and extracellular matrix without a doubt play a fundamental role in tissue mechanical homeostasis, our platform is explicitly rigged to study such dynamics.

### 2.2. Pathophysiological Framework

Blood vessels are responsive to mechanical stresses, facilitating long-term growth and remodelling by adaptation of geometry, structure, and material properties [27]. The two major mechanical stresses are wall shear stress and wall stress, with the former related to blood flow and the latter to blood pressure. It is important to note that wall shear stress and wall stress are physically and physiologically very different entities and are therefore not to be confused.

A core concept we propose is that any circumstance, chronic or transient, that may derange mechanical homeostasis, has the potential to elicit a vicious cycle into pathophysiology.

Accordingly, the sensing and actuating roles of endothelial cells and VSMCs should be considered, whilst pathophysiological triggers at these levels are very likely to derange mechanical homeostasis. As such, this serves as a framework for the study of aTAA formation, further detailed below.

#### 2.2.1. Wall Shear Stress Homeostasis

Systemic organ and tissue perfusion requires unimpeded flow through blood vessels, such as the aorta, which translates into frictional forces at the endothelial interface: *wall shear stress*. Informed by the endothelium as a sensor, the regulatory “rule” of the vessel is to keep wall shear stress constant in the face of changes in blood flow: when wall shear stress increases due to a long-term increase in flow, a structural dilation will bring wall shear stress back to a normal level; and vice versa if flow decreases [28]. In patient-based aortic aneurysm research, wall shear stress cannot be directly measured and hence is estimated from computational modelling of the dynamic in- and outflow through the vessel, with local geometry and dimensions obtained by dedicated medical imaging [29]. Wall shear stress is indeed clearly implicated in aTAA formation, and 4D-flow MRI and computational fluid mechanics techniques have advanced such characteristics in the patient context [30].

So far, we have assigned the endothelium the role of the sensor in the above regulatory loop, but we still need to consider the actuator as well. Aortic and conduit artery dilation and constriction are mediated by around 50 up to 80 concentric layers of VSMCs, forming lamellar units. In these lamellar units, VSMCs are intricately sandwiched as a single-cell-thick layer between elastin laminae reinforced by interwoven collagen fibres [31]. The regulation of wall shear stress involves paracrine signalling (cell–cell communication) to VSMCs through relaxants/constrictors. Dependent on a corresponding relaxation/contraction response of the VSMCs, the desired vessel dilation/constriction occurs [28]. Hence, the functional status of the VSMCs (signalling sensitivity, tone, ability to contract) and corresponding lamellar unit (ECM stiffness) are of direct relevance to endothelium-mediated modulation of aortic diameter and *wall shear stress homeostasis.*

#### 2.2.2. Wall Stress Homeostasis

Wall stress is defined as the tensional force carried by the vessel wall across its wall area. Blood pressure (more precisely transmural hydrostatic pressure) is a clinically relevant hemodynamic variable triggering adaptive responses of vessel wall thickness and geometry to achieve *wall stress homeostasis* [26,32,33,34].

Wall stress homeostasis involves a complex network of interrelated processes, such as mechano-sensing [35], VSMC contractile responses [26,36,37], actin–myosin complex and cytoskeletal arrangements and remodelling [26,38], cell–cell communication, extracellular matrix (ECM) remodelling, and VSMC phenotypic modulation [39]. With VSMC contractile responses we refer to myogenic, cell–cell related, as well as neurogenic contractile responses of VSMCs. It is relevant to note that the evidence for myogenic response in large arteries and the aorta is scarce, in contrast to the body of experimental data on arterioles and smaller resistance arteries where this phenomenon has been clearly implicated in normal physiology as well as in disease-related remodelling [36]. The abovementioned multi-level possibilities leave open a significant number of research questions around the regulation of wall stress and its derangement in aTAA.

Given the relevance of cell–matrix interactions, particularly between VSMCs and ECM, to both wall stress and wall shear stress homeostasis, our main research question is: **How is wall stress regulated through VSMC activity, in interaction with the ECM and endothelium, and how is it deranged in aTAA(D) patients?**

### 2.3. Focus of MAPEX Platform Infrastructure

Due to the associated multitude and hierarchy of regulatory levels and consequent complexity, wall stress homeostasis currently remains a key area of research interest. In relation to the (putative) role of VSMCs in mechanical homeostasis, their phenotypic diversity and modulation are clearly important to consider. Correspondingly, the latter two aspects are priorities in our research. In addition, our platform deliberately facilitates the assessment of local differences in cell–matrix biology and biomechanics, rendering patients their own control and allowing for paired statistical analyses. Accordingly, we have implemented the following features into MAPEX:Preoperative 4D-flow MRI, to capture wall shear stress patterns in relation to local vessel geometry;Strain imaging during open-chest surgery, to capture local vessel wall deformations;Tissue sampling co-localised with local tissue strain and wall shear stress measurements;To capture tissue and cell-specific biomarkers, as well as ultrastructural properties;Inclusion of a large cohort (minimal target of 400) of aneurysm patients undergoing cardiac surgery, complemented by patients with normal aortic diameter as controls (with limited tissue sampling);Data structuring ready for (i) computational biomechanical modelling, (ii) omics, and (iii) artificial intelligence approaches, for data integration and interpretation, as well as new hypothesis generation.

By integrating the above scientific and infrastructural elements, our MAPEX platform is well-positioned to innovate our understanding of aTAA formation at a fundamental level, with good potential for translational prognostic and therapeutic approaches.

## 3. Methodological Setup

### 3.1. Acquisition Platform Management

MAPEX captures a prospective observational surgical patient population, spearheaded by the departments of cardiothoracic surgery, biomechanical engineering, and biochemistry. The platform also includes subjects with (paediatric) congenital heart disease associated with aortic dilatation.

The medical ethics committee of Maastricht University Medical Center exempted the acquisition platform/biobank from the scope of the Dutch Medical Research Involving Human Subjects Act (reference: METC-2019-1235). The collection, storage, and processing of patient data and materials are in accordance with the “Code for Proper Use of Secondary Human Tissue in the Netherlands” and conform to the principles outlined in the Declaration of Seoul.

### 3.2. Study Population

Patients undergoing any type of aortic repair with an open-chest approach are recruited at Maastricht University Medical Center (MUMC+) (Figure 1). This includes patients with ascending, root, or arch aneurysms, as well as patients with an acute type A dissection. In addition, patients undergoing coronary artery bypass grafting (CABG) or aortic valve replacement (AVR) without aortic aneurysms are included as a control group. Patients with congenital heart disease, including paediatric patients, are recruited particularly at University Medical Center Groningen (UMCG) expert centre. All clinical variables such as age, sex, height, weight, diagnosis, US/CT/MR imaging, and other clinical data are collected during the pre- and peri-operative phases. MAPEX considers a longitudinal cohort of minimally 400 subjects, aiming to include 70 to 80 subjects each year.

### 3.3. Recruitment Procedure

All eligible subjects are identified at the outpatient or inpatient clinics of the departments of Cardiothoracic Surgery at the MUMC+ and UMCG. Eligible patients are informed about the goal of the research project, the implications, values, and possible risks. All patients included in the acquisition platform have given written informed consent. No patients were invited to contribute or to comment on the study design or interpret the results. For the paediatric population (age < 18) informed consent has been given by parents or legal representatives. For the dissection patients, written consent was obtained from either the patient or legal representative at a later stage post-operatively.

All clinical data are collected from the local hospital information systems. Clinical data such as patient information, medical history, risk factors, clinical variables, and medication use (illustrated in Table 1) are collected during ongoing patient care. The clinical database is implemented using Castor EDC (Ciwit, Amsterdam, The Netherlands, which is hosted by True21 in the Netherlands) and hosted on a server in a secure and controlled environment. These systems are compliant with Good Clinical Practice (GCP) and are managed by ISO27001 certified data centres. All patients enrolled in this study are being registered under a participant study number to protect patient data transparency. This study number is used for inclusion, data collection, and data processing.

### 3.4. Pre- and Peri-Operative Procedures

*Pre-operative phase* (Figure 2)*:* All patients with aTAAs scheduled for surgery and eligible for MR imaging, undergo preoperative 4D-flow MRI screening of the total aorta on a 3T MR system (Philips Ingenia; Philips Healthcare, Best, The Netherlands). The MRI protocol includes coronal and axial cine images of the left ventricular outflow tract (LVOT) and cross-sectional cine images of the aortic valve. Also, anatomic images of the thoracic aorta are achieved during diastole using retrospective ECG-gating and navigator respiratory gating based on diaphragm excursion. Furthermore, two-dimensional phase contrast images are acquired at multiple levels along the aortic length (ascending aorta, descending aorta, thoraco-abdominal aorta), with a temporal resolution of 5–7 ms, and VENC adjusted for flow velocities at the respective level. Three-dimensional and three-directional phase contrast imaging, also known as 4D-flow MRI, are performed for the total thoracic aorta with a spatial resolution of 2.5 × 2.5 × 2.5 mm and a temporal resolution of 30–40 ms. Images are acquired using retrospective ECG-gating and navigator respiratory gating based on diaphragm excursion. Post-processing of 4D-flow data is performed using commercially available software (CAAS MR Solutions 5.2.1; Pie Medical Imaging, Maastricht, The Netherlands). The lumen of the aorta is segmented from the 4D-flow data set. Regional flow-related parameters such as flow rate, flow velocity, and wall shear stress are extracted from the CAAS MR solutions software [40].

*Peri-operative phase* (Figure 2)*:* After sternotomy, intra-operative video-based strain measurement of the ascending aorta is performed to estimate biaxial strains for local wall deformation [41]. Briefly, the videos are made during both “low” and “high pressure” conditions through neutral, anti-Trendelenburg and Trendelenburg positions, with blood pressure recorded from the hemodynamic monitor. Four sterile surgical pledgets (BARD PTFE Felt Pledgets, Bard Peripheral Vascular, Inc., Tempe, AZ, USA), are sutured to the exposed adventitial epiaortic fat of the ascending aortic aneurysm around the maximum diameter as assessed by the surgeon. The pledgets define four different strain measurement locations namely proximal and distal transverse, and medial and lateral longitudinal. Videos of the ascending aorta are obtained using a GoPro camera (HERO7, GoPro Inc., San Mateo, CA, USA) thereafter. These pledgets serve as markers and are tracked in all the frames of the video using a local processing algorithm to determine marker positions (Figure 2). Using the marker positions, the distances between each marker pair are determined, thereby giving estimates of time-varying local circumferential and axial dimensions. Beat-to-beat varying circumferential and axial dimensions ultimately enable the calculation of (dimensionless) local biaxial strains of the ascending aorta [41].

During the perioperative phase, two types of biomaterials are collected: blood and residual tissue.

Non-heparinised blood samples are collected from the central line catheter in an EDTA BD Vacutainer ^®^ and sodium citrate BD Vacutainer^®^ tubes (Table 2). Blood is processed, for serum, EDTA plasma, citrated plasma (for coagulation assay), and buffy coat (peripheral blood mononuclear cells (PBMCs) for induced pluripotent stem cell (iPSC) generation) isolations. These are then stored for analysis and further research. Note: the human iPSC generation is currently performed in selected cases.

During the aneurysm repair procedure, the surgeon decides on the aneurysm tissue segment to be resected (which includes the pledget strain markers). The resected tissue is mapped along the coordinates provided by the preoperative MRI grid (Figure 3) and then further sectioned into four different anatomical segments aorta ventral (AV), aorta medial (AM), aorta lateral (AL) and aorta dorsal (AD). For each segment, tissue samples are stored directly (1) in RNA later stabilisation solution, (2) in liquid nitrogen as snap-frozen tissue, (3) in 4% paraformaldehyde for paraffin-embedded sections (FFPE), (4) embedded in O.C.T. compound medium for fresh-frozen sections, and (5) in serum-free M199 of DMEM—without FCS and 1% penicillin/streptomycin (Table 2). The latter tissue sample is transferred to cell culture facilities for immediate processing to obtain primary VSMCs for cell culture. All biomaterials are coded with each participant number and stored at the local aTAA tissue bank within MUMC+. All procedures are performed using the standard operating procedures (SOPs). Additionally, all collected biomaterials are registered and documented in a “Tissue registration form”.

## 4. Materials and Data Processing

### 4.1. Location-Specific Characterisation Methods of Aortic Tissue

#### 4.1.1. Aortic Tissue Samples

Tissues of each region of the ascending aorta (Aorta Ventral AV; Aorta Dorsal AD; Aorta Lateral AL; Aorta Medial AM) are fixed in 1% paraformaldehyde for paraffin-embedded specimens (FFPE); see Table 2 and Figure 4. Series of 4 μm thick transverse sections are obtained from paraffin-embedded tissue samples for histological analysis.

#### 4.1.2. Histopathological Analysis

Haematoxylin and eosin (HE) staining is used to assess aortic wall morphology, media nuclei count and inflammatory changes, elastin Verhoeff–van Gieson (EVG) stain to examine elastin content and fibre thickness, and Picro Sirius Red stain to examine collagen content in the aortic wall (Table 3). SMC contractile proteins (calponin; CNN-1 and alpha-smooth muscle actin; αSMA) were calculated from antibody staining using rabbit anti-calponin antibody and mouse anti-αSMA (Table 3 and Figure 4). The positive pixel % of CNN-1 and αSMA were calculated over the total surface area (μm^2^) for each anatomical segment. The slides are screened using VENTANA iScan HT slide scanner microscope or Olympus TH4-200 BX61VS at a magnification of 10 or 20×. The relative positive staining per area surface is quantified using bioimage analysis software QuPath version 0.3.2. To investigate the impact of wall shear stress and/or strain gradients on the connective fibre microstructure of each region in ascending aorta, tissue data will be correlated with the regional flow-related parameters, such as flow rate, flow velocity, and wall shear stress of 4D-flow MRI and to the local wall deformation analysis obtained from the video tracking method using the geometric mapping grid (Figure 2 and Figure 3).

#### 4.1.3. Genotyping and Omics-Approach

TAAD is known to involve degeneration of the medial layer/lamellar units. We aim to gain, from an omics approach, insights into the processes of aortic wall degeneration and potentially rupture susceptibility. Multiomic profiling of the genome, transcriptome, and proteome of TAA tissue, together with non-aneurysmatic samples will be performed using our MAPEX biobank in discovery analyses.

Whole genomic sequencing will provide insight into genetic variation in patients with TAA wherein structural and functional changes in tissue can be linked. Potential applications of epigenetic-based analysis, such as DNA methylation, acetylation or ATAC-based sequencing, can be used with genomic and transcriptomic analysis to gain further insight into the mechanisms governing pathological processes. Transcriptomics will be applied to both tissue, primary cell isolates and iPSC-derived cells. This will allow us to appreciate the downstream impacts of genetic dysregulation in TAA. Proteomics and metabolomics by a mass spectrometry approach will be key to identifying the nature of structural and metabolic dysregulation (Figure 4). This approach will allow us to profile molecular features at the individual cell level and allow us to correlate molecular findings with clinical measurements such as aortic diameter, and global and regional tissue deformations, as well as flow-related parameters derived from 4D-flow MRI (Figure 3). Combined, these detailed analyses will help generate novel insights into potential sub-populations presenting with aTAAD.

### 4.2. Circulating Markers and iPSC-Oriented Research

#### 4.2.1. Circulating Blood Markers

Blood collected as serum EDTA and citrated plasma are stored as part of the biobank generation for various assays. These serum or plasma samples may be of value in the determination and validation of singular biomarkers such as DES and IDES involved in aTAA progression (Table 3). Further, these can be used to culture vascular smooth muscle cells and determine possible modulation of cell phenotype [33]. Together with tissue analysis, measuring the expression of circulating markers will determine not only how expression changes but also help us to develop a predictive tool for the diagnosis (Figure 4).

#### 4.2.2. Pluripotent Stem Cell Research

Isolation of PBMCs allows us to generate induced pluripotent stem cells (iPSCs), embryonic-like cell lines that can be expanded infinitely, differentiated into vascular cells, and are patient-specific (Figure 4) [42]. All patients included in iPSCs research are screened for pathogenic gene variants conferring a high risk for heritable aTAA using a TAAD gene panel (Table 4). Of great interest for aTAA research, we have optimised differentiation protocols for the generation of lineage-specific VSMCs conferring to smooth muscle cell subpopulations that predominantly reside in either ascending aorta, aortic arch, or descending aorta (Figure 5) [43]. This allows us to generate an unlimited supply of patient-specific VSMCs from different aortic loci to model aortopathy [44]. Thus far, we have generated a single iPSC line from an aTAA patient with Marfan. This line is called CARIMi004 and is registered on the hPSCreg^®^ (https://hpscreg.eu/cell-line/CARIMi004-A, accessed on 20 July 2023). We aim to generate new cell lines from both syndromal (Marfan, Loeys–Dietz, Ehlers–Danlos) as well as non-syndromal aTAA patients in the coming years. In an ongoing research project, CELLSYSTEMICS, we will develop individual patient-oriented approaches to characterise and understand the role of VSMCs in wall stress and wall shear stress homeostasis (https://www.health-holland.com/project/2022/2022/how-get-cells-talking, accessed on 20 July 2023).

Concomitant expansion of stem cell research with the MAPEX platform will allow us to validate and expand on findings from biomechanical and clinical data. Direct comparison of transcriptomic, and proteomic profiles between iPSC-VSMCs and primary cells will enable further insight into the nature of the disease and be of benefit to stem cell research and translation [45]. Integration of iPSC-VSMCs with 3D and dynamic mechanical systems (stretch and indentation) could open mechanistic insights into SMC mechanobiology in relation to pathology and has potential application in next-generation personalised drug screening platforms that can aid in personalised treatment strategies [46].

## 5. Statistics and Methodology

The accumulated and growing data will be used to address a range of main and underpinning research questions around aTAA(D) pathophysiology at different levels: genetic—omic—cell—tissue—vessel—patient. With the targeted substantial number of subjects, we will be able to apply multivariate analysis and AI approaches for testing hypotheses.

In addition, computational biomechanical and physiological models will be used to integrate appropriate data to help interpret cell–matrix biology findings. Using variability estimates as input, these models are also used to simulate virtual patient populations, allowing (re)classification of actual patients.

Because of the integrative multidisciplinary approach beyond aortic morphology, MAPEX will also allow the exploration of new markers by hypothesis generation through machine learning algorithms.

## 6. Research Scope

We have adopted and developed a multidisciplinary view on aTAA formation, resulting in the MAPEX research platform (Figure 6). Our core translational focus is to capture and explain the variety in pathology in our aTAA(D) patient cohort. Considering the unexplained variations between patients and aTAA pathology, we aim to translate these towards heterogeneity and diversity at the cell–matrix level, particularly the dynamic interactions between VSMCs and ECM. Here, we introduce some of our current ideas and aims, defining the research scope of MAPEX.

Arterial wall remodelling traits such as dilatation, stiffening, and calcification are governed by multiple signalling pathways, wherein homeostatic crosstalk between VSMCs and ECM plays a key role. Early signs of ECM degradation, including elastin degradation and fibrillin fragmentation by MMPs, can be monitored by DES and ISDES in blood plasma or urine. Other (potentially) relevant plasma biomarkers are TLR-4 [47], MMPs [48], FBN1 [49], SEF [50], TGF-β [51], PDGF-B [52], ACE/ACE2 [53], α-2-HS glycoprotein [54] and dp-ucMGP [55]. We aim to investigate these biomarkers through MAPEX, because their potential use in clinical practice appears very attractive.

As stated above (Section 2), the regulatory role of VSMCs in wall stress homeostasis as well as their phenotypic plasticity appear yet overlooked. To capture the functional aspects of VSMC phenotype, which are directly relevant to mechanobiological responses [32], we will focus on studies comparing primary VSMCs obtained from aTAA wall tissues (“disease cells”) and “naïve cells”, generated from PBMCs/human-induced pluripotent stem cells (hiPSC) differentiated into VSMCs.

Mechanobiological cues about (loss of) mechanical homeostasis, we believe, come from measuring the local strain differences across the aTAA [41]. The aortic bend and associated morphology of the developing aneurysm brings up questions about what physical deformations one may expect but especially about how VSMCs and other cells may respond to differences in strain. Would they realign? Would they contract? Would they reorder? Would they remain silent?

By computational modelling imaging-based geometry and video strains, we aim to derive patient- and location-specific estimates of aortic wall elastic properties and mechanical stress [56], as experienced and potentially governed by VSMCs. From this thread, we hope to learn about how VSMCs facilitate wall stress homeostasis, and what triggers or conditions derail such regulatory processes.

Further in line with our physiological framework, we will consider the associations of (differences in) wall shear stress with (corresponding differences in) observed histological and cell biology findings. Other studies have investigated the association of wall shear stress with histology with some clear outcomes, implicating elevated wall shear stress as a hemodynamic driver of tissue degradation [30]. Our work will provide further mechanistic insight into these associations, by due consideration of the role of VSMCs as actuators in wall shear stress homeostasis. Such advances in understanding may support better prognosis and monitoring of aTAA.

Ultimately, the MAPEX platform—through its direct patient context—is well-suited to identify novel screening or prognostic biomarkers as well as help develop cell-based regenerative therapies.

## 7. Clinical Application

The MAPEX platform approach will allow us to profile mechanistic features at the cell tissue level and correlate these with existing clinical measurements, such as aortic diameter, and global and regional tissue deformation, as well as flow-related parameters derived from 4D-flow MRI.

As a multicentre registry and because of its bio- and data-banking nature the MAPEX platform will provide opportunities for implementing multivariate statistics and AI to assess complex multi-scale analyses of the pathophysiological mechanisms.

The integration of iPSC-derived VSMCs with 3D and dynamic mechanobiological phenotyping methods may offer patient-specific mechanistic insight into disease variants. Such insight has great potential in personalised screening programs to support and improve patient risk management/prognosis and cell-based/regenerative treatment strategies.

## 8. Future Directions

The MAPEX platform brings an innovative integration of scientific approaches and methodologies in the field of ascending thoracic aortic disease. The combination of deep phenotyping at histological and cell biology levels with imaging-based wall shear stress and wall strain measurements is a key innovative strength of our MAPEX platform.

The integrative interpretation of such data in the direct patient context has real potential of narrowing existing translational gaps and may ultimately improve the clinical management of (a)symptomatic patients with existing and developing thoracic aortic disease.

## Figures and Tables

**Figure 1 biomedicines-11-02095-f001:**
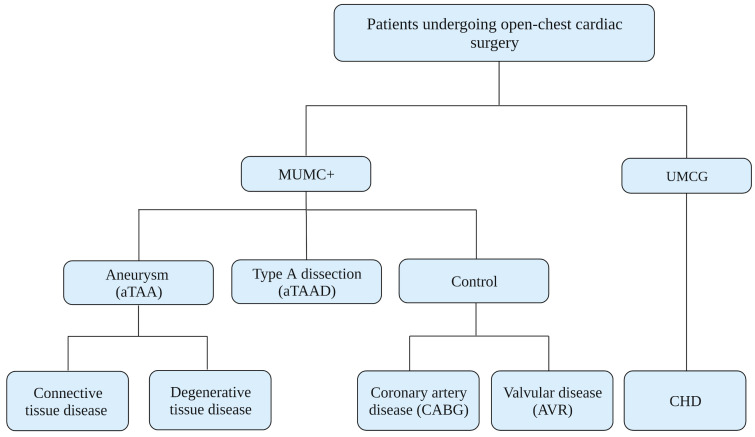
Current patient sources. aTAA = ascending thoracic aortic aneurysm; aTAAD = aTAA dissection; CABG = coronary artery bypass grafting; AVR = aortic valve replacement; CHD = congenital heart disease. Clinical centres: Maastricht University Medical Center (MUMC+), University Medical Center Groningen (UMCG).

**Figure 2 biomedicines-11-02095-f002:**
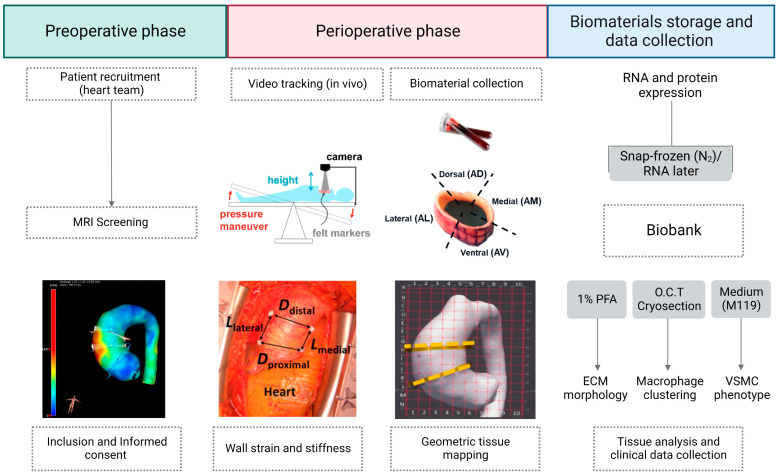
Overview and timing of MAPEX data and material collection, processing, and storage.

**Figure 3 biomedicines-11-02095-f003:**
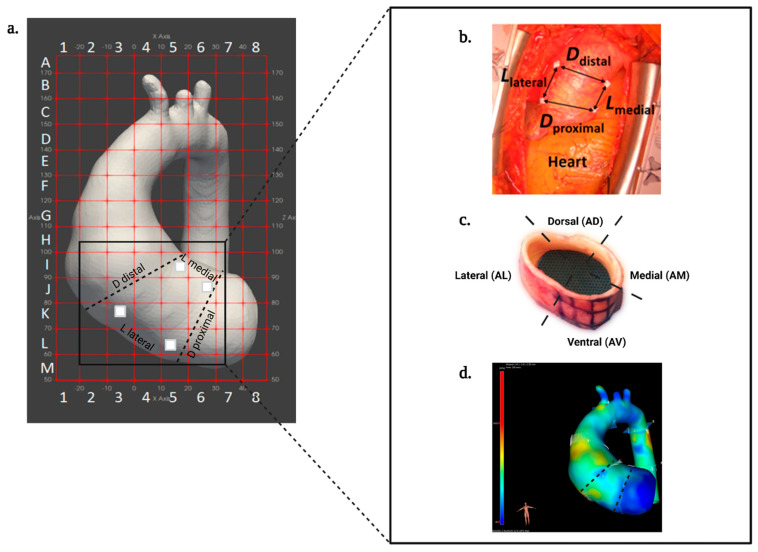
Cross-modality mapping of patient-specific biaxial strains, tissue sampling, and wall shear stress, by utilising systolic pre-operative 4D-flow MRI anatomy. (**a**) MRI-grid for mapping, (**b**) intra-operative video tracking markers, (**c**) resected aortic ring with 4 main anatomical orientations ventral (AV), medial (AM), lateral (AL), dorsal (AD), and (**d**) estimation of 4D wall shear stress field.

**Figure 4 biomedicines-11-02095-f004:**
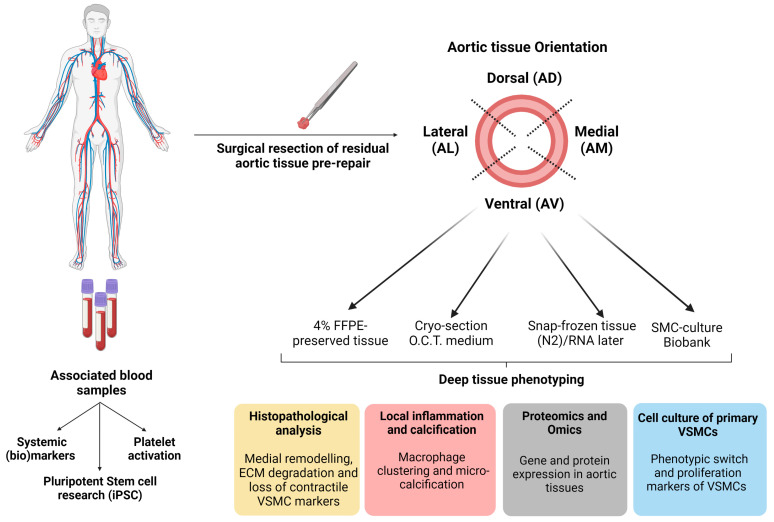
Location-specific characterisation of ascending aortic aneurysm tissue as well as systemic arterial blood, enabling correlation to local differences in mechanical stresses.

**Figure 5 biomedicines-11-02095-f005:**
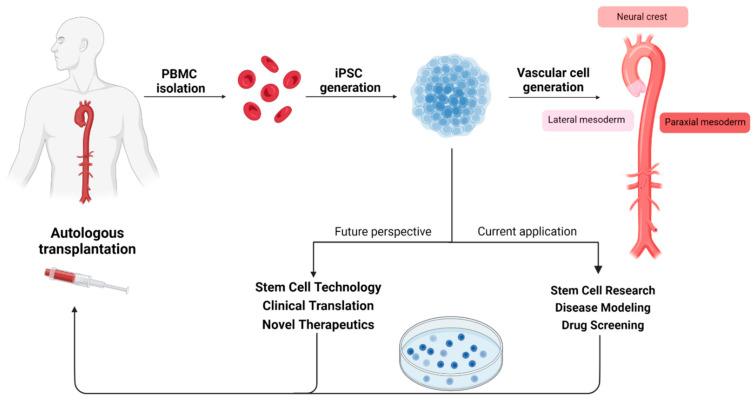
MAPEX-related perspective on human induced pluripotent stem cell (hiPSC) based research and potential applications.

**Figure 6 biomedicines-11-02095-f006:**
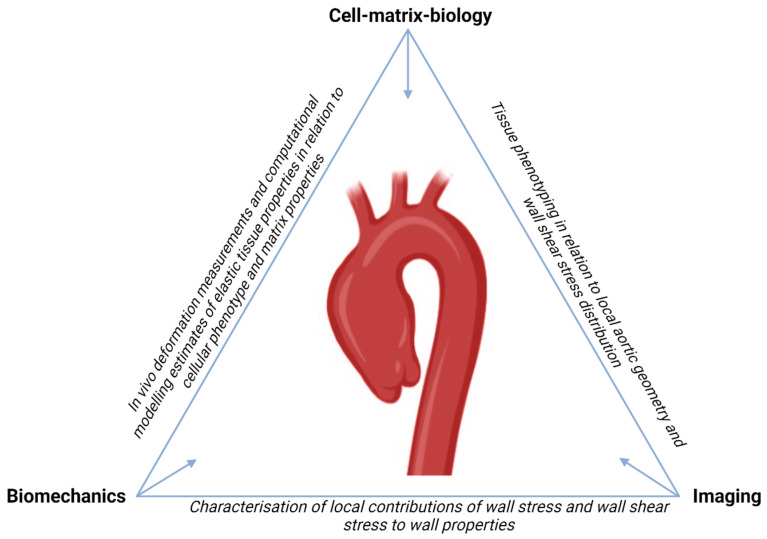
MAPEX triangle of interdisciplinary research focus on the pathophysiology of aTAA formation.

**Table 1 biomedicines-11-02095-t001:** Clinical characteristics of current cohort.

	aTAA	HTAA	Controls	aTAAD
Type of index event	Aneurysm	Heritable aneurysm	CABG/AVR	Type-A dissection
Number of subjects	86	9	30	20
Gender (male %)	70	67	90	65
Age (years)	60 ± 14	44 ± 14	63 ± 12	60 ± 14
BMI (kg/m^3^)	26 ± 5	23 ± 4	26 ± 4	28 ± 7
Weight (kg)	81 ± 17	67 ± 12	83 ± 16	85 ± 24
Aortic diameter (mm)	53 [47;59]	48 [40;60]	41 [36;44]	55 [52;52]
Hypertension (%)	73	33	77	95
Diabetes mellitus (%)	5	11	7	-
Hypercholesterolemia (%)	53	-	70	60
COPD (%)	7	22	10	20
Myocardial infarction (%)	9	-	23	20
Family history for CVD (%)	20	56	53	20
Current smoker (%)	9	22	27	25
Alcohol use (%)	65	56	57	45
Aortic insufficiency (%)	77	67	40	35
Aortic stenosis (%)	20	-	60	5
Valve morphology (% BAV)	31	44	27	10
LVEF (%)	53 [46;59]	55 [52;57]	55 [45;60]	-

Table shows status of the MAPEX database from October 2019 till April 2023. Age, BMI, and weight are presented as mean +/− SD and other values as median with IQR [Q1–Q3]. aTAA = ascending thoracic aortic aneurysm; aTAAD = aTAA dissection; HTAAD = heritable thoracic aortic aneurysm. CABG = coronary artery bypass grafting; AVR = aortic valve replacement; BAV = bicuspid aortic valve; TAV = tricuspid aortic valve.

**Table 2 biomedicines-11-02095-t002:** Patient biomaterials collection and storage.

Biomaterials	Storage	Analysis Material	Storage Temperature
Blood	EDTA BD Vacutainer	Plasma	<−80 °C
		PBMCs	<−80 °C
	Sodium Citrated BD Vacutainer	Serum	<−80 °C
Tissue	Serum free M199/DMEM medium	Tissue sections for smooth muscle cell isolation	4 °C
	4% paraformaldehyde-fixed	Paraffin-embedded tissue sections	20–22 °C
	Fresh frozen in OCT	Fresh-frozen tissue sections	<−80 °C
	RNA later	RNA stabilised tissue for gene expression	<−80 °C
	Snap frozen in N_2_	Snap-frozen tissue for gene expression	<−80 °C

EDTA = ethylene diamine tetra acetic acid; PBMC = peripheral blood mononuclear cell; M199 = Medium 199; DMEM = Dulbecco’s modified eagle medium.

**Table 3 biomedicines-11-02095-t003:** Type of material analysis and markers of interest.

Category	Phenotypes	Marker of Target	Type of Analysis	Name of Stain/Assay	References
Tissue	Cell nuclei, extracellular matrix, cytoplasm, inflammatory infiltration, adipose tissue	VSMC nuclei	Histological	Haematoxylin and eosin	SOP 13LU-0406
	Extracellular matrix	Collagen I and III	Histological	Picrosirius red (Direct red 80)	SOP 16CM-421
	Extracellular matrix	Elastin	Histological	Elastica van Gieson	SOP 12PL-0403
	Contractile VSMC	Calponin-1 (CNN1)	Immunohistological (IHC)	CNN1 stain	SOP 14LU-0412
	Contractile VSMC	Alpha-smooth muscle actin (α-SMA)	Immunohistological (IHC)	α-SMA stain	SOP 14LU-0404
	Vascular vitamin K status and vascular calcification	Uncarboxylated matrix gamma carboxyglutamate protein (ucMGP)	Immunohistological (IHC)	ucMGP stain	SOP 14LU-0413
	Proteinase inhibitor	Alpha-1 Antitrypsin (A1AT, SERPINA1)	Immunohistological (IHC)	SERPINA1 strain	SOP 22RK BG-439
Blood serum	Breakdown products of elastin	Desmosine (DES)	Enzyme-linked immunosorbent assay (ELISA)	DES ELISA kit	n.a.
	Breakdown products of elastin	Isodesmosine (IDES)	Enzyme-linked immunosorbent assay (ELISA)	IDES ELISA kit	n.a.

SOP = standard operating procedure, as locally established and managed.

**Table 4 biomedicines-11-02095-t004:** Composition of the TAAD gene panel for thoracic aortic aneurysm and dissection.

*Gene*	*Transcript Reference* *(Ensembl)*	*Alternative Exon*	*Transcript Reference for Alternative Exon* *(Ensembl)*
*ABL1*	*ENST00000372348*		
*ACTA2*	*ENST00000458208*		
*ARIH1*	*ENST00000379887*		
*BGN*	*ENST00000331595*		
*COL3A1*	*ENST00000304636*		
*EFEMP2/FBLN4*	*ENST00000307998*		
*ELN*	*ENST00000358929*		
*EMILIN1*	*ENST00000380320*		
*FBN1*	*ENST00000316623*		
*FBN2*	*ENST00000262464*		
*FLNA*	*ENST00000369850*		
*FOXE3 ^a^*	*ENST00000335071*		
*HCN4*	*ENST00000261917*		
*IPO8*	*ENST00000256079*		
*LMOD1*	*ENST00000367288*		
*LOX*	*ENST00000231004*		
*LTBP3*	*ENST00000301873*		
*MAT2A*	*ENST00000306434*		
*MFAP5*	*ENST00000359478*		
*MYH11*	*ENST00000452625*	*Exon 42B*	*ENST00000396324*
*MYLK*	*ENST00000360304*		
*NOTCH1*	*ENST00000277541*		
*NPR3*	*ENST00000265074*		
*PLOD1*	*ENST00000196061*	*Exon 2A*	*ENST00000449038*
*PMEPA1/TMEPAI*	*ENST00000341744*		
*PRKG1 ^b^*	*ENST00000401604*		
*ROBO4*	*ENST00000306534*		
*SKI*	*ENST00000378536*		
*SLC2A10*	*ENST00000359271*		
*SMAD2*	*ENST00000402690*		
*SMAD3*	*ENST00000327367*	*Exon 1A*	*ENST00000439724*
*SMAD4*	*ENST00000342988*		
*SMAD6*	*ENST00000288840*		
*TGFB2*	*ENST00000366929*		
*TGFB3*	*ENST00000238682*		
*TGFBR*	*ENST00000374994*		
*TGFBR2*	*ENST00000359013*		
*THSD4*	*ENST00000261862*		

^a^ Forkhead domain only; ^b^ Exon 3 only. For the complete TAAD gene panel, a coverage of 20× for >99% of the coding sequence is postulated. For the following core genes, coverage of 100% is guaranteed: *ACTA2*, *COL3A1*, *FBN1*, *FLNA*, *MAT2A*, *MFAP5*, *MYH11*, *MYLK*, *NOTCH1*, *PRKG1*, *SMAD2*, *SMAD3*, *TGFB2*, *TGFB3*, *TGFBR1* and *TGFBR2.* Large deletions/duplications are not detected with the current version of the assay. All annotations are based on the Hg19/GRCh37 genome build. Classification of detected sequence variants is performed according to the 5-class system: non-pathogenic (class 1, polymorphism), probably non-pathogenic (class 2), clinical significance unclear (class 3), likely pathogenic (class 4) and pathogenic (class 5, mutation).

## Data Availability

The data presented in this study are available on request from the corresponding author. The data are not publicly available due to privacy considerations.

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
