# Peer review of "The Maastricht Acquisition Platform for Studying Mechanisms of Cell–Matrix Crosstalk (MAPEX): An Interdisciplinary and Systems Approach towards Understanding Thoracic Aortic Disease"

_biomedicines, 2023, doi:10.3390/biomedicines11082095_

Round 1

Reviewer 1 Report

Dear authors,

I appreciate your work and I believe this study brings a valuable contribution in improving diagnostic and prognostic possibilities to identify and treat patients with existing and developing aneurysms. 

I have a few minor things to point out:

The description of Figure 1 is too long. I believe it should be better to explain in the main text all the details and discussion reffered to figure 1 and try to keep a briefer description.

Please clarify the use of subsections 6.1, 6.2 and 6.3. If they are indded subsections of chapter 6, please define a title for them. If not, please remove the numbers.

English language needs minor improvements

Author Response

Dear Reviewer, 

Please find attached our responses to the points you and other reviewers raised. 

Kind regards,

The authors. 

Reviewer 2 Report

Authors provide report on the design, structure, capabilities, and status of the MAPEX. The cohort of patients whose data and bio-samples have been collected (2019-2023) and included in the MAPEX database is also briefly described. The study is relevant as the population of patients with thoracic aortic disease is heterogeneous and risk prediction in borderline aortic dilatation is challenging. The introduction is well written and relevant. Impaired VSMC contractility may indeed play a role in aortic aneurysm pathophysiology though data confirming this hypothesis are scarce. 

The Section 2 has some flaws. 

Page 4, Fig. 1.  Authors present the figure showing “myogenic tone” as one of the factors of wall. However, ‘myogenic tone’ and ‘myogenic response’ is a very specific physiological phenomenon present only in the resistance arteries. It is a hall-mark feature of resistance arteries and their downstream arterioles. I strongly recommend using a different term instead of “myogenic tone” in regard to the aorta.

Page 4, line 133. Fig. 1 and its caption are hardly comprehensible, and the figure requires to be redrawn for better clarity of the expressed ideas. What “heterogeneity” and “local biaxial gradients” do Authors refer to in the Fig. 1?

Page 5, line 168. Authors refer to “myogenic response [31, 35]” while discussing aortic wall stress homeostasis. However, the references [31, 35] do not discuss myogenic response in the context of aorta at all. Neither myogenic tone nor myogenic response is present in the aorta. I recommend using a different term instead of “myogenic response” in regard to the aorta.

While discussing Fig. 1, authors cite the Ref. [28], but this paper does not present data on the aorta at all. I recommend Authors citing papers focusing on the aortic studies instead of referring to works on other arteries such as carotid, brachial, and superficial femoral arteries like in Ref. [28]. The aorta has its own unique biomechanical characteristics, pathophysiology, and morphology distinct from those of smaller blood vessels even the conduit ones.

Page 6, line 226. Abbreviation “UMCG” should be explained in the Fig. 2 caption.

TAAD gene panel is presented in Table 4.

Regarding future directions: Perhaps epigenetic modification such as DNA methylation may be also studied using the MAPEX and accumulated biosamples. DNA methylation study could generate valuable data from both mechanistic and biomarker perspectives. Genome-wide DNA methylation analysis may contribute to better clustering of patients. Cross-tissue correlations of genome-wide DNA methylation may be considered. 

Once again, my main suggestion for the Authors is to cite (and consider in the future) papers presenting data of human aortic studies. Papers focusing on smaller human arteries or animal models may be irrelevant and could be potentially misleading hampering breakthrough in aTAA research. 

Author Response

(The authors gave the same response as above.)

Reviewer 3 Report

In this excellent review paper, the authors describe the research methodology and the finality of MAPEX , an interdisciplinary platform that aims to extend the knowledge on aortic disease. 

The paper is well written and the matter of interest 

In order to increase the appeal of the  paper I suggest adding a section on the clinical applicability of the expected result

The paper is well written

Author Response

(The authors gave the same response as above.)

Round 2

Reviewer 2 Report

Dear Authors, 

Thank you for the discussion and addressing the comments. The revised version of the manuscript has been improved. The project MAPEX you present  is highly appreciated.